# Pathways to Acceptance in Participants of Advanced Cancer Online Support Groups

**DOI:** 10.3390/medicina57111168

**Published:** 2021-10-27

**Authors:** Christina Francesca Pereira, Kate Cheung, Elyse Alie, Jiahui Wong, Mary Jane Esplen, Yvonne W. Leung

**Affiliations:** 1de Souza Institute, Toronto, ON M5T 1V4, Canada; christinaf.pereira@mail.utoronto.ca (C.F.P.); kate.cheung@mail.utoronto.ca (K.C.); elyse.alie@mail.utoronto.ca (E.A.); Jiahui.Wong@desouzainstitute.com (J.W.); 2Department of Psychiatry, Faculty or Medicine, University of Toronto, Toronto, ON M5S 1A1, Canada; maryjane.esplen@utoronto.ca

**Keywords:** acceptance, advanced cancer, mindfulness, distress, existential suffering, social support, palliative care

## Abstract

*Background and Objectives*: Individuals with cancer, especially advanced cancer, are faced with numerous difficulties associated with the disease, including an earlier death than expected. Those who are able to confront and accept the hardships associated with the disease in a way that aligns with their beliefs benefit from more positive psychological outcomes compared to those who are aware of their diagnosis but are unable to accept it. To date, there is limited research exploring factors contributing to illness and death acceptance in the context of advanced cancer in group therapy settings. *Materials and Methods*: The current study used a Directed Content Analysis approach on transcripts of online advanced cancer support groups to investigate if and how Yalom’s existential factors played a role in the emergence of acceptance. *Results*: The online support group platform, combined with the help of facilitators, offered supportive environments for individuals seeking help with cancer-related distress by helping patients move towards acceptance. Some participants had already begun the process of accepting their diagnosis before joining the group, others developed acceptance during the group process, while a few continued to be distressed. Our analysis revealed the emergence of four themes related to illness acceptance: (1) Facilitator-Initiated Discussion, including sub-themes of Mindfulness, Relaxation and Imagery, Changing Ways of Thinking, and Spirituality; (2) Personal attitudes, including sub-themes of Optimism and Letting Go of Control; (3) Supportive Environment, including the sub-themes of Providing Support to Others and Receiving Support from Others; and (4) Existential Experience, which included sub-themes of Living with the Diagnosis for an Extended Amount of Time, Legacy and Death Preparations, and Appreciating life. *Conclusions*: With a paradigm shift to online delivery of psychological services, recognizing factors that contribute to acceptance when dealing with advanced cancer may help inform clinical practices. Future studies should explore patient acceptance longitudinally to inform whether it emerges progressively, which has been suggested by Kübler-Ross.

## 1. Introduction

With recent medical advances in personalized medicine, people with advanced or metastatic cancer are now living longer and higher quality lives than before [1]. However, the diagnosis of an advanced cancer may still elicit feelings of emotional distress for patients [2], which is why the development of coping strategies, such as acceptance, are important for increasing patient well-being and reducing psychosocial distress [3]. Generally, acceptance can be conceptualized as the ability to remain present and available to current experiences without trying to change the unpleasant or bask in the pleasant [4]. More specifically, in the context of a terminal illness, acceptance is considered to be an active approach in which a person does not attempt to judge or change the course of their illness or their internal experience [5]. Rather, the person chooses to take a self-compassionate approach towards these feelings in a way that aligns with their core values and maintains their self worth, while still acknowledging the struggles of the illness, including the inevitability of their mortality [3,6]. Nevertheless, there is a distinction to be made between disease acceptance and resignation, or fatalism, which refers to when a person remains hopeless and passive about their situation [7]. Thus, there is a balance between accepting what one cannot change (more active coping) while not succumbing to resignation, which is a clinically desirable outcome that can often be worked towards with psychosocial support and interventions [7].

Another acceptance-related concept that is especially therapeutic for individuals faced with advanced cancer is accepting the terminality of one’s life, also known as death acceptance. Individuals that are aware of their poor prognosis but have not come to accept or are still fearful of their impending death have significantly more distress [8]. If patients with advanced cancer are able to confront the diagnosis and accept its finality, they endorse more beneficial outcomes, such as feeling less anxious and distressed [9], feeling more peaceful [10], and have improved quality of life [11]. Given the benefits of illness and death acceptance among cancer patients, it is essential to recognize the factors that contribute to these attitudes so that psychosocial services can facilitate and encourage acceptance among people with different beliefs, cognitive styles, and personalities.

The recent public health restrictions due to the COVID-19 pandemic have served as an impetus for digital transformation to address mental health needs virtually, and as a consequence, digital means have now become the main mode of mental health service delivery [12]. The de Souza Institute offers CancerChatCanada (CCC), a series of synchronized, therapist-led, text-based Online Support Groups (OSGs) for cancer patients and caregivers. CCC is a national program that is in collaboration with six provincial cancer agencies in Canada. The OSGs vary in number of sessions, purpose, and model, with all groups being manual-based and consisting of 8 to 10 sessions [13]. During the sessions, the facilitators aim to support the group by promoting discussion based on the sessions’ themes and any related concerns while also acknowledging and attending to the members’ emotional needs. The facilitators are well-versed in employing group therapeutic factors, grounded in Yalom’s theory of group psychotherapy [14], which they use to promote mutual support among the group members. Along with session-specific themes, facilitators also guide members through sets of homework readings and interactive activities. During the group discussion, facilitators may instill a wide range of therapeutic factors originally proposed by Yalom, including altruism, cohesion, universality, interpersonal learning, guidance, catharsis, identification, instillation of hope, and existential factors surrounding death [14]. Yalom argued that humans must recognize the pain and unjustness of life and death. This is a realization that individuals must come to on their own in order to have acceptance. It can be hard to come to terms with these difficult truths; therefore, group therapy can create a facilitating environment for individuals to face these facts. The group therapeutic process can be used to gently guide people to this realization through discussions and the aforementioned therapeutic factors, but ultimately, it is the responsibility of the individual to accept his or her existential circumstances. To date, few studies have examined how acceptance emerges in group therapy in relation to these therapeutic factors. Does acceptance emerge organically, or does the group facilitator play a role in promoting acceptance? Further research is needed to understand the process of group support and how it facilitates this important therapeutic outcome, especially in the age of virtual care.

### Objectives

One aim of this study was to investigate whether Yalom’s existential factors associated with illness-related acceptance and, potentially, additional therapeutic factors (e.g., organic discussion or other patient factors) could be identified in therapist-led OSGs. A second aim is to understand how cancer patients express their illness-related acceptance using a qualitative research approach.

## 2. Materials and Methods

### 2.1. Participants

Individuals from CCC, an OSG service offered by the de Souza Institute, consented to having their chat history recorded and analyzed. Consent was obtained through an online form prior to engaging in the OSGs. During the consenting process, group members were infomred that participation was voluntary. If they wished to refuse consent or withdraw their consent, they were able to join other OSG sessions that were not subject to analysis. This ensured that group members did not feel that they needed to participate in order to receive the psychosocial services they were seeking out.

This dataset was ideal, as CCC offers synchronous, text-based OSGs that are led by professionals to facilitate discussion and promote mutual support among members [13]. To ensure participant anonymity, all identifying information mentioned in the sessions, such as names and locations, were masked using asterisks before analysis. The data from two advanced cancer support groups were qualitatively coded by human scorers to extract information regarding how group members came to accept their illness. The support groups provided an encouraging and safe environment for individuals to share information related to their advanced cancer diagnoses. Discussion was led by facilitators and guided by specific discussion topics from the NuCare Manual [15]. In total, there were 18 participants (See Table 1 for participant characteristics). Participants’ ages ranged from 25 to 65+, with the majority being between 55 and 64. Most of the patients were female (94.4%) and had either breast (61.1%) or colorectal (22.2%) cancer. All participants were advanced cancer patients. This analysis is part of a larger study using applications of artificial intelligence (AI) to enhance virtual care, which was approved by the University Health Network Research Ethics Board and conducted under the ethical standards highlighted in the Declaration of Helsinki.

Transcripts were analyzed by three undergraduate research assistants (C.F.P., K.C., E.A.) who were supervised by a Ph.D.-level scientist (Y.W.L). The team adopted the Directed Content Analysis approach [16,17] to identify pathways taken to achieve illness- and death-related acceptance within the therapy sessions, based on the theory of existential factors by Yalom [14]. The existential theory posits that humans must face that the mortality, freedom, and responsibility for their own life are rooted in isolation, aloneness, and the lack of intrinsic meaning. In the context of life-limiting diseases, such as terminal cancer, the acceptance of fear and decrease in quality in life is needed to experience the fullness of life [14]. Therefore, the group facilitators attempted to orient the discussion to promote the development of existential factors by encouraging the expression of emotions, facilitating vicarious learning among group members, and validating their experiences in the context of advanced disease.

### 2.2. Data Analysis

A coding guide (Appendix A) on acceptance-related literature was developed to deductively identify how participants expressed their acceptance in a consistent manner. By viewing these internal experiences through a compassionate and non-judgemental lens, it allows one to reevaluate group members’ experience in a way that is consistent with their core values. The initial coding guide was developed based on Yalom’s [14] notion about existential awareness and acceptance as well as other acceptance-based literature [3,5,6,18,19,20].

### 2.3. Directed Content Analysis

A combination of deductive and inductive techniques were used to determine our coding strategy and analyze the comments made by participants [21]. Previously developed collections of theories of acceptance [3,5,6,14,18,19,20] allowed us to initially deduce instances during which participants were expressing accepting attitudes towards their advanced cancer diagnoses, death, and dying. Inductive techniques were used during the more in-depth analysis of the text to identify themes and sub-themes. Units of coding included levels of single-sentence, single-message, single-session, and trans-session.

This process involved undertaking the following steps:

First, the entire transcripts from the sessions were repeatedly read to become familiarized with the topics of conversations. Second, the Acceptance Coding Guide was used to determine if individuals were engaging in acceptance speech or not. Instances that aligned with the developed criteria of the coding guide were highlighted. Third, we assessed acceptance at the single sentence, message, and session levels; then, quotes of each participant across therapy sessions were examined to understand how they described their acceptance. Fourth, the highlights were further refined to establish new categories and subcategories of codes in an inductive manner to create a more in-depth understanding of what factors led to participants’ acceptance during the course of group support. Finally, observations were made for each example of acceptance to describe what led to the comment being said during the session, and if the member indicated any factors that contributed to the development of these attitudes. After each coder had completed their initial assessment of the transcripts, meetings were held to assess the reliability of the individual findings, and discrepancies were resolved. Thus, these meetings ensured consistency and allowed for facilitated conversations about the content to further refine the codes.

## 3. Results

Table 2 displays the quotes, themes, and sub-themes of acceptance that were extracted by the team. There were four themes established, each with multiple sub-themes. We identified techniques, which were based on Yalom’s theory of existential factors, that were adopted by group facilitators to guide the group’s discussion. These techniques included facilitating universality, altruism, interpersonal learning, cohesion, catharsis, and instillation of hope. Throughout the sessions, participants openly shared their personal stories and expressed their concerns and fears about advanced cancer. Aside from the discussions of the assigned readings, participants showed immense support to each other in the face of advanced disease. Death- and illness-related acceptance emerged naturally in many contexts as well as in response to facilitator-led topics. Indicators of acceptance were associated with facilitator-initiated prompts, personal attitudes, environmental factors, and life experiences related to existential issues. Figure 1 provides a theory of the pathway to acceptance based on the themes and sub-themes that emerged from the dataset. Each theme is detailed below with relevant quotes capturing participants’ acceptance. Personally-identifiable information has been concealed from the quotes using asterisks. 

### 3.1. Theme 1: “Facilitator-Initiated Discussion”

Using the assigned readings, the facilitators initiated conversations on the following topics to instill Yalom’s therapeutic factors, such as the installation of hope, universality, catharsis, altruism, and existential factors, in group members to help them face the harsh truths of advanced cancer. Facilitators created a safe environment to encourage the open expression of existential concerns and personal stories as well as displayed empathy, encouragement of affect, reflective listening, and validation of participants’ subjective experience. For example, one facilitator acknowledged feelings of loneliness and isolation: “*Wanting to gather us together in our group folks… and just send out a wave of support and care… hearing the isolation of being alone right now*.” Facilitators also promoted a safe space where participants did not have to always be positive, but could express all emotions: “*In this group, I think there is a fine balance--between focusing on coping alongside allowing the full range of emotions… grief, loss, and sadness. Please let’s be gentle with ourselves and not feel like we can only bring our positive experiences and feelings here*.” Finally, they encouraged social support: “*I wonder if it would be possible for just a moment- to really experience the support of this group folks… just noticing that you are right here with other people that get your experience and your fears.*” Prior to each session, the group facilitator assigned participants a reading from the NuCare Manual for Mixed Diagnosis [15]. The manual facilitated discussion and aided in allowing participants to express their reactions to readings and their acceptance attitudes related to the following sub-themes.

#### 3.1.1. Sub-Theme 1: Mindfulness

Members were introduced to the coping strategy of mindfulness, described as a state of awareness towards the present moment without emotionally reacting to it [22]. During the discussion, the facilitator seamlessly transitioned the conversation from members’ biggest concerns to promoting Yalom’s factor of installation of hope [14] by acknowledging members’ “*moments of acceptance*” and “*accepting that this is what is*”. At the beginning of the session, a participant acknowledged the difficulties of the illness while still maintaining their sense of self: “*I try to be mindful daily. It can be hard, but I just really try to enjoy each moment and savour it. I’m so happy to be alive. I read some of the coping skills, and I like the saying ‘the present moment is the only time that any of us ever has’*”. Later in the group session, the facilitator summarized the session by saying “*so many good coping tools folks- walking even in the winter, connecting with friends and family, staying in the moment, letting go of too much planning, meditating, yoga, and accepting*”, which altered the conversations towards accepting aspects of their illness/life that they cannot control. One participant responded “*going through diagnosis, treatment, etc., taught me to let go of control/planning*”, while another group member stated, “*I was a control freak. Not anymore. I’ve let it all go… and it almost is freeing*.”

#### 3.1.2. Sub-Theme 2: Relaxation and Imagery

Group members were invited to practice allowing themselves “*to relax to the degree that they choose*” and to “*clear the mind of negative thoughts*”, which is essential to practicing imagery, another coping strategy. The facilitator guided the OSG through an imagery activity where group members imagined being on a beach to foster relaxation and a sense of calmness. Upon reflecting on the activity, one participant commented on the hardships of the disease while relishing in the beneficial feelings of support by the group and the need for relaxation: “*Mostly that my feelings are shared by others. I am not alone or isolated in my feelings. This disease sucks. Bad. We need to give ourselves moments of self calm. Even briefly*”. This statement highlighted how the group discussion and facilitator direction were able to elicit feelings of universality among the group. Shortly after, another member followed with: “*While I’m aware of my limitations, the power of the mind and being able to take ourselves away from our reality can make a difference-for me, anyway. Even for a brief amount of time*.” These conversations allowed the participants to come to terms with and accept that they may not always be able to do things physically, but relaxation and imagery can help achieve a different dimension of possibilities by providing them with a sense of hope. Reminding OSG members about the power of their mind allowed some to verbally acknowledge the shared limitations associated with the disease (universality) but also that these limitations need not consume every aspect of their lives (instillation of hope) [14].

#### 3.1.3. Sub-Theme 3: Changing Ways of Thinking

This discussion revolved around how thoughts about facts can impact feelings and explored test-related anxiety, feelings of grief and loss, and corrected negative self-talk. The therapist prompted the group by asking how the members “*shift [their] ways of thinking around cancer*?” One member responded, “*I accept it right away. I have never been angry or asked why me. Why not me! Better that I have it and not my mom or sister. Then I am thankful that there are treatments for me, and I just carry on. It is my new normal and I am still thankful*”. Later in the session, the conversations evolved as another member asked how “*others train [their] brain*?”, to which the same group member that responded to the therapist offered, “*We all have choices *****. Most day (sic) I choose to find the good. Everyday I find something good. Start with simple things. Focus on others and not me*.” During the session, the discussion shifted from discussing coping strategies to discussing negative emotions that the members may be feeling. This allowed the members to share their grief and losses associated with the disease. A group member acknowledged the feelings related to the death of a friend, but conceded, “*I try to live in the moment and appreciate the little things instead of wallowing in grief*”, displaying that, while they understood that grief was part of the process, they were still able to maintain appreciation of their life. This conversation topic was influential in the emergence of acceptance and the process of group therapy in general because the group was able to display examples of Yalom’s therapeutic factors of altruism and catharsis [14].

#### 3.1.4. Sub-Theme 4: Spirituality

The weekly discussion topic was about practicing a healthy lifestyle, which included a focus on spirituality and how that fostered acceptance of illness and death. The facilitators focused on religious and non-religious spirituality, then skillfully transitioned the conversation to an open discussion about death, thereby incorporating Yalom’s existential therapeutic factor into the session. The group facilitator started by asking about how cancer impacted members’ relationship with spirituality. One participant commented *“Faith is important to me. And it helps me [cope]”.* After some comments about how cancer increased the importance of spirituality in some participants’ lives, the facilitator then asked participants, “*if part of faith is thinking more about death and what happens after we die?*” While some participants voiced their concerns, others expressed having “*a good relationship with death*”, and one group member went as far to say: “*[death] does not scare me at all […] I have had a near death [experience] so I have no fear left […] No fear at all actually. It was one of the most profound moments of my life.*” This facilitator-led discussion invited participants to share their acceptance with the group to help comfort other members’ death-related concerns and aided in stimulating the interpersonal learning in group interventions.

### 3.2. Theme 2: “Personal Attitudes”

Expressing personal attitudes that members believed contributed to their overall well-being allowed for the potential of the therapeutic factor of interpersonal learning to take place among other members. The following personal attitudes or traits contributed to the organic experience of acceptance of some individuals.

#### 3.2.1. Sub-Theme 1: Optimism

The sub-theme of optimism emerged when certain group members endorsed having a positive outlook. Some participants expressed that they had “*always been optimistic*”, “*tend to be a positive thinker*”, or expressed go-with-the-flow attitude when facing advanced disease. For example, one topic of discussion was longing for a return to a pre-cancer normal, when a member tried to bring positivity and acceptance to the conversation by saying, “*This is our new normal, unfortunately. But it does not have to be all bad and negative. I find positivity every day. I have to. It really helps me”*.

#### 3.2.2. Sub-Theme 2: Letting Go of Control

Another noteworthy sub-theme was related to members’ ability to let go of control. Many participants expressed these sentiments over the conversation about returning to normalcy. One participant commented that they had accepted that they needed to take things slower while recognizing having to change their ways of thinking: “*I have a 15-year-old daughter with autism. I stay positive for her. I also take one day at a time now. Which was hard as I was a chronic ‘planner’ person. lol. But taking things step by step now keeps me more positive as well as reaching out to others.”* Another member expressed the same feeling of letting go while talking about having to balance pre-cancer responsibilities with the cancer-related fatigue they were feeling: “*I always try to push through and be the hero… that I can do this… and I do love my kitchen, cooking, etc… but I bit the bullet and wow… I feel so free knowing its (sic) one less thing I have to worry about”*.

### 3.3. Theme 3: “Supportive Environment”

Feeling loved and supported from others played a large role in the acceptance of terminal illness. Concepts related to having a socially supportive environment were raised by group members in two distinct ways.

#### 3.3.1. Sub-Theme 1: Providing Support for Others

The group provided a sense of group cohesion, or togetherness, and altruism to the participants, even finding comfort and increasing their own acceptance by supporting others within the group or recounting stories of when they had supported non-group members. For example, a participant acknowledged that cancer was “*scary though[t] but we are in it together*” and another mentioned, “*I always find comfort in supporting others who are going through the same thing”*.

#### 3.3.2. Sub-Theme 2: Receiving Support from Others

Numerous participants also described that they were able to accept their condition because of the support they have received. For example, one member stated, “*I have great support from my immediate family and friends*,” while another said, “******-I cope with it by going outside and enjoying nature. Spending as much time with my kids as well*.” When discussing death, a participant gave emotional and informational support to another member while also expressing death acceptance: “*it has been somewhat helpful for me to look into some support around death. It may be very early but it helps to feel like I have my ducks in a row. I’ve looked into Greensleeves, MAID [Medical Assistance In Dying] etc*.” These discussions helped promote the factors of altruism, cohesion, and even universality among the group members.

### 3.4. Theme 4: “Existential Experience”

Learning of a cancer diagnosis can bring many existential themes to the forefront of people’s lives. Many of the participants endorsed existentially-related concepts when expressing their attitudes of acceptance, such as the following sub-themes.

#### 3.4.1. Sub-Theme 1: Living with Diagnosis for an Extended Amount of Time 

Participants described that they accepted their illness because they have lived with it for an extended period of time. A participant expressed: *“I never got angry when first diagnosed 11 years ago. I am still not angry as it won’t do me any good. This is what I have been dealt with and so this is what I have to do. And I would rather have this than my mom or sister. I can do this and I am.”*

#### 3.4.2. Sub-Theme 2: Legacy and Death Preparations

Another existential-related sub-theme emerged when an individual initiated a conversation on legacy. One member was able to recognize death as an inevitable outcome that everyone faces and explained: “*I have taken over someone else’s legacy and am adding to it and then will pass it on to the next person when the time comes*.” This comment’s sentiment of passing the legacy on expressed the inevitable threat of death and their acceptance of this.

#### 3.4.3. Sub-Theme 3: Appreciating Life

It seemed like for some, losing someone to cancer allowed them to appreciate life, and that led to acceptance of cancer of their own. For example, during initial introductions, one participant stated, “[I] *will always be in treatment but thankful and happy to be here. Side effects and all*” while another said, “*I am thankful that I am still here enjoying life. I just had another friend lose her battle on wednesday (sic) and it makes me appreciate life so much!*”. These individuals expressed that despite all the negatives associated with their diagnoses, they are still accepting of the life they have, including the illness.

## 4. Discussion

The objective of this current study was to uncover pathways to acceptance of a terminal illness from the chat history of advanced cancer OSG members. A Directed Content Analysis approach guided by Yalom’s [14] theory of group psychotherapy was used on CCC transcripts. We also looked for additional factors to acceptance among these group members. The results are summarized in four themes: Facilitator Initiated Discussion, Personal Attitudes, Supportive Environment, and Existential Experiences. These themes inter-played and contributed to the acceptance of advanced disease and death acceptance. Our findings suggested that being older, e.g., 55 years old or older as opposed to being 54 years old or younger, was also slightly associated with expressing more acceptance. The findings also showed that group facilitators successfully applied and promoted many of Yalom’s therapeutic factors through the sessions, such as cohesion, catharsis, installation of hope, universality, altruism, interpersonal learning, and existential factors [14]. Facilitators used these skills to encourage the discussion of existential issues by providing weekly readings and inviting the group members to openly discuss their experiences and support each other, which in turn, promoted cohesion. The assigned readings supported altruism and interpersonal learning among participants, as they openly shared with the group about their coping strategies, including the use of mindfulness, imagery, and goal setting. The weekly readings also facilitated catharsis through the acknowledgement and expression of members’ negative emotions, including fear, grief, and loss. The facilitators made an explicit effort to acknowledge successes of group members, an example of the instillation of hope, and to highlight the universality between members’ feelings and situations. Acceptance happened as a result of the group discussion of the weekly readings and also happened organically as participants shared about their life experiences, personalities, and world views, and encouraged and supported each other.

The themes discovered about acceptance herein are supported by the literature. For example, personality research has shown that individuals with more optimistically oriented personalities seem to be willing to confront their crisis and engage in growth opportunities, motivated by achieving life goals [23]. Their impending deaths bring these individuals to an enhanced focus on what matters in their life, such as living authentically and searching for spirituality and meaning, all of which allows them to come to terms with their death [23]. The sub-theme of “spirituality” is a broader concept than religiosity, referring to the personal experience of expressing ideas and attitudes and understanding life, like when searching for meaning in a disease [24]. Mindfulness training often includes a large component of spirituality. Spirituality is associated with reduced psychological distress, as people who are more spiritual are more able to perceive positive outcomes from the cancer experience, which contributes to less reported symptoms [25]. Existential experiences are also supported by literature that states trauma often leads to an increased spirituality in some survivors, which may help them find personal meaning in the traumatic experience [26]. This concept is referred to as post-traumatic growth and can increase the spiritual acceptance of illness and death. However, contradicting the literature that suggested that younger female breast cancer patients (under 55) exhibited greater degree of disease acceptance [27], we found that older patients (above 55) were more likely to express acceptance. As the stage of disease may play a role, future studies are needed to tease out the complexity of age as a factor influencing acceptance in cancer patients.

Our findings suggest that acceptance in advanced cancer patients in the group program occurred through group process and the emerging topics being discussed, participants’ personalities, perceived supportive environment, spirituality, existential experiences, and desire to help others, which is supported by the work by Yalom [14]. We theorize that the facilitated discussions employing group process techniques and readings from the NuCare Manual are reflection triggers, which, when delivered in a socially supportive environment to individuals with certain personal attitudes and past existential experiences, facilitate and enhance the emergence of acceptance. Thus, some participants may be more likely to expereience acceptance if they possess certain personal attitutes and existential experiences (Figure 1). Undoubtedly, OSGs are important in providing a platform for advanced cancer patients to safely interact and come to terms with their advanced diagnosis [14]. This interaction exhibits aspects of humanistic and existential approaches, including the display of empathy, encouragement of affect, reflective listening, and acceptance of others’ subjective experience [28]. In some cases, rather than imposed by the facilitator, participants applied these existential therapy techniques naturally to reach acceptance of advanced cancer.

While there was an abundance of acceptance related to members’ illnesses, there was relatively less death-related acceptance. There was some discussion regarding death and dying throughout the group sessions, with some examples of death acceptance, as one member mentioned: “*living my best life until I die.*” However, explicit death acceptance was scarce, with only a few members speaking directly and openly about it. Despite the lack of overt comments accepting the finality of their lives, most of the members were able to admit that they had taken steps to prepare for death. Comments about preparing wills, discussing end-of-life preparations, and sorting through their possessions gave the impression that they were aware and had acknowledged the proximity of their death despite not wanting to talk about the actual event itself. One potential explanation for this is that these individuals are taking a more neutral approach; they are not being overly introspective about their feelings towards death, but they are dealing with it in a “matter of fact” way and preparing for it physically. Another explanation that was brought up during the sessions is that society often denies death-related conversations or finds the subject to be unmentionable: *“I wonder folks what makes it harder to talk about a terminal illness? Is it because our society denies death?”* Therefore, instead of feeling comfortable enough to discuss the event of their death, these individuals could only mention the preparations they were taking out of necessity. Indeed, the facilitators were skilled at creating a safe environment to discuss their biggest fears and concerns, seemingly in hopes of cultivating death acceptance, and reminded certain group members that their cancer was not just a chronic condition but that it was incurable. The confrontation of this notion that society denies death may be useful for therapists to cultivate more group conversations surrounding death and lead to more acceptance. Further, other models that are longer term in nature, such as the supportive expressive group therapy model, have demonstrated patterns of participant expressions around death in a more open and direct way in populations of advanced cancer [29].

Nevertheless, these findings that individuals were reluctant to discuss death openly suggest that OSGs may be better suited to support death acceptance in individuals who are already beginning to accept it, but may not be the best avenue to facilitate death acceptance in those who are still quite distressed or are not ready to accept their death. Given the existential nature of accepting the end of one’s life, perhaps more intensive, targeted, and personalized interventions are needed to explore such a difficult topic for some people. In fact, the nature of an online setting could be a deterrent for facilitating in-depth conversations, as when attempting to cultivate authenticity and bonding, since there is a lack of visual feedback, such as body language and other subtle non-verbal cues. Moreover, group facilitators may feel challenged in an online group model, as they may have more limited capacity for cultivating the well-known therapeutic factors [14], as there are multiple participants with various issues in the group conversing with text simultaneously. Future work should explore ways to enhance the online therapist’s ability to facilitate existential factors in virtual care. Our findings support the literature that online therapists could adopt exercises, such as mindfulness and the use of a gratitude diary, in the context of existential discussion to enhance acceptance [30].

### 4.1. Limitations

There are some limitations of the current study that should be mentioned. Firstly, just because participants did not express their experiences associated with acceptance, that does not mean that they had no experiences of acceptance at all. We could only detect acceptance based on their discussion in the text transcripts of group sessions. Therefore, future works should actively explore acceptance using surveys or qualitative interviews with group participants in addition to the written transcripts. This study is also limited by the lack of formal follow-up procedure for dropout participants. This prevented us from potentially drawing conclusions about how the emergence of acceptance progresses through time, as our final sample only included information on individuals who completed the OSG sessions and therefore, may be biased. Future work should implement a formal follow-up procedure to determine why some individuals stopped their participation, which can inform how these participants may be different from those who complete the therapy program. Another limitation of this study is that it was only conducted on a small group of individuals from a Western culture, specifically Canada; thus, results may not be generalizable to other parts of the world. Future studies should examine how acceptance may emerge in different cultures as expression of cancer-related distress and acceptance may not be as openly discussed and accepted as it is in the West [31]. The sample size also limited our ability to link certain demographic factors, such as age and sex, to the expression of illness and death acceptance.

### 4.2. Future Directions

There may have been additional underlying factors that contributed to participants’ expression of acceptance that can be assessed with longitudinal studies. Future work in this field should investigate how the stages of grief proposed by Kübler-Ross may be involved in death acceptance in advanced cancer patients [32]. The theory of Kübler-Ross shares some similarities to that of Yalom, but it also proposes some alternative explanations regarding the emergence of acceptance. While both Kübler-Ross and Yalom identify denial as a deterrent from acceptance, their theories differ, however, regarding how individuals can overcome this denial. Instead of relying on therapeutic factors to help individuals achieve acceptance, Kübler-Ross’s theory focuses on the sequential progression through stages, such as denial, anger, bargaining, depression, and finally acceptance. There is limited research to support one theory over the other; therefore, future studies could investigate if advanced-care patients progress through these stages and how this process may be aided by the guidance of facilitators. This may be achieved by a longitudinal study that collects repeated measurements of participants’ experiences as they progress through therapy sessions using multiple interviews or self-report questionnaires. This future work may further elucidate how illness- and death-related acceptance emerges in OSGs.

## 5. Conclusions

Acceptance in advanced cancer patients involves the interaction between therapist-facilitated topics, personality, a supportive environment, and their existential views. Advanced cancer patients did not frequently express their experience of death acceptance within the group discussion; however, when they did, they did it to support each other. Future studies should explore ways to facilitate the process of these interactions within OSGs and investigate how individuals’ accepting attitudes emerge over time using longitudinal analyses to enhance the quality of OSGs.

## Figures and Tables

**Figure 1 medicina-57-01168-f001:**
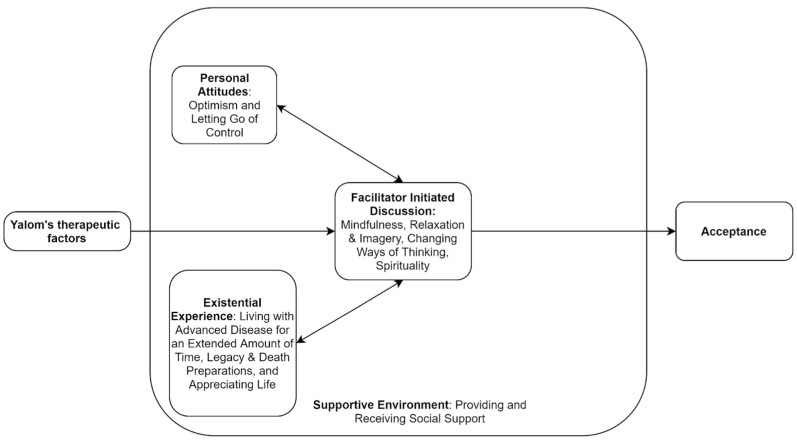
The summary and relationship between themes and sub-themes that contributed to the emergence of acceptance in Online Support Group (OSG) participants.

**Table 1 medicina-57-01168-t001:** Demographic information of participants (*n* = 18). Includes data related to gender, age, location, type of cancer, and treatment status.

Characteristics	Value (*n* = 18), *n* (%)
Gender	
Female	17 (94.44)
Male	1 (5.55)
Unknown	0 (0)
Age group (years)	
18–24	0 (0)
25–34	1 (5.55)
35–44	4 (22.22)
45–54	2 (11.11)
55–64	10 (55.55)
65+	1 (5.55)
Location	
British Columbia	6 (33.33)
Ontario	7 (38.88)
Alberta	2 (11.11)
Other province	3 (16.66)
Type of cancer	
Breast	11 (61.11)
Gynecological	0 (0)
Colorectal	4 (22.22)
Head and neck	1 (5.55)
Other cancers	1(5.55)
Unknown	1 (5.55)
Treatment status	
Active treatment	0 (0)
Post treatment	0 (0)
Other	18 (100)

**Table 2 medicina-57-01168-t002:** Quotes from the Online Support Groups (OSG) indicating acceptance of participants, organized into themes and sub-themes. Quotes are labeled by session date, user ID, gender, and age.

Themes	Sub-Themes	Quotes from the Patients (Session, Participant ID Number, Sex, Age)								
Facilitator-initiated discussion	Mindfulness	“I try to be mindful daily. It can be hard but I just really try to enjoy each moment and savour it. I’m so happy to be alive. I read some of the coping skills, and I like the saying ‘the present moment is the only time that any of us ever has’” (Advanced_Cancer_Session1_30 April 2021; User ID: 3100, female, 35–44)	“I have always been optimistic-not (sic) everyone is. I purposely find the good in people and in life. We really don’t have to look too far in our everyday lives to find that we are in fact very lucky to be living here in Canada.” (Advanced_Cancer_Sesson1_30 April 2021; User ID: 3212, female, 55–64)	“I think going through diagnosis, treatment etc (sic) taught me to let go of control/planning.” (Advanced_Cancer_Session1_30 April 2021; User ID: 3034, female, 55–64)						
	Relaxation and Imagery	“While I’m aware of my limitations, the power of the mind and being able to take ourselves away from our currently reality can make a difference-for (sic) me, anyway. Even for a brief amount of time. Can help a bit with the boredom too. lol.” (Advanced_Cancer_Session2_7 May 2021; User ID: 2338, female, 35–44)	“Mostly that my feelings are shared by others. I am not alone or isolated in my feelings. This disease sucks. Bad. We need to give ourselves moments of self calm. Even briefly.” (Advanced_Cancer_Session2_7 May 2021; User ID: 3197, male, 45–54)							
	Ways of thinking (grief and loss, changing their cancer thoughts,	“I try to Live in the moment and appreciate the little things instead of wallowing in grief.” (Advanced_Cancer_Session3_14 May 2021; User ID: 3208, female, 35–44)	“The rewards of staying in the moment and recognizing and changing negative self talk are so stress reducing.” (Advanced_Cancer_Session3_14 May 2021; User ID: 3034, female, 55–64)	“I accept it right away. I have never been angry or asked why me. Why not me! Better that I have it and not my mom or sister. Then, i am thankful that there are treatments for me, and I just carry on. It is my new normal, and I am still thankful.” (Advanced_Cancer_Session3_14 May 2021; User ID: 3212, female, 55–64)						
	Spirituality (religious, non-religious, and death)	“Faith is important to me. And it helps me” (Advanced_Cancer_Session7_11 June 2021; User ID: 3212, female, 55–64)	“***** I cope with it by going outside and enjoying nature. Spending as much time with my kids as well.” (Advanced_Cancer_Sesson1_30 April 2021; User ID: 3100, female, 35–44)	“Actually laughter and spirituality can be linked […] Not taking yourself so seriously, enjoying life, living in the moment.” (Advanced_Cancer_Session7_11 June 2021; User ID: 3203, female, 55–64)	“We are [a death-denying society], but [death] does not scare me at all […] I have had a near-death experience so I have no fear left […] No fear at all actually. It was one of the most profound moments of my life.” (Advanced_Cancer_Session7_11 June 2021; User ID: 3203, female, 55–64)	“i have a good relationship with death” (Advanced_Cancer_Session7_11 June 2021; User ID: 3203, female, 55–64)	“Death does not bother me a lot.” (Advanced_Cancer_Ssession7_11 June 2021; User ID: 2987, female 65+)			
Personal attitudes	Optimism	“We all have choices ****. Most day (sic) I choose to find the good. Everyday I find something good. Start with simple things. Focus on others and not me.” (Advanced_Cancer_Session3_14 May 2021; User ID: 3212, female, 55–64)	“** I was also scared about the word “Palliative” and the added team on my already overwhelming list. But I have quickly realized that palliative can also be years and years of continued life.” (Advanced_Cancer_Session1_30 April 2021; User ID: 3215, female, 25–34)	“This is our new normal, unfortunately. But it does not have to be all bad and negative. I find positivity every day. I have to. It really helps me.” (Advanced_Cancer_Session1_30 April 2021; User ID: 3212, female, 55–64)						
	Letting Go of Control	“Me tooI (sic) was always planning and controlling things but now I Have totally let go.” (Advanced_Cancer_Sesson1_30 April 2021; User ID: 3100, female, 35–44)	“I have a 15 year old daughter with autism. I stay positive for her. I also take one day at a time now. Which was hard as I was a chronic “planner” person. lol. But taking things step by step now keeps me more positive as well as reaching out to others.” (Advanced_Cancer_Sesson1_30 April 2021; User ID: 2338, female, 35–44)	“i always try to push through and be the hero… that I can do this… and i do love my kitchen, cooking etc… but I bit the bullet and wow… I feel so free knowing its (sic) one less thing I have to worry about.” (Advanced_Cancer_Session5_28 May 2021; User ID: 3215, female, 25–34)	“I find that my lists allow me to keep track of things and I feel less stressed as I often forget things.” (Advanced_Cancer_Session5_15 July 2020; User ID: 2557, female, 55–64)					
Supportive environment	Receiving Support from Others	“Thanks! I wish that everyone finds some peace and hope to enjoy each days (sic) as it comes…, Thanks everyone!.,Ha ha ha ha ha ha ha ha.,Feeling very supported!” (Advanced_Cancer_Session7_29 July 2020; User ID: 2417, female, 55–64)	“Feeling very supported!.,Bye everybody! Have a good week!”(Advanced_Cancer_Session6_22 July 2020; User ID: 2417, female, 55–64)	“MoStly that my feelings are shared by others. I am not alone or isolated in my feelings. This disease sucks. Bad. We need to give ourselves moments of self calm. Even briefly.” (Advanced_Cancer_Session2_7 May 2021; User ID: 3197, male, 45–54)	“***** - I cope with it by going outside and enjoying nature. Spending as much time with my kids as well.” (Advanced_Cancer_Sesson1_30 April 2021; User ID: 3100, female, 35–44)	“I have great support from my immediate family and friends.” (Advanced_Cancer_Session5_28 May 2021; User ID: 3215, female, 25–34)	“Yes I am surrounded by love and support. That’s one reason that I can help support others.” (Advanced_Cancer_Session8_18 June 2021; User ID: 3212, female, 55–64)	“I find talking to my close friends and sister very helpful to ease the thoughts… chocolate yes!” (Advanced_Cancer_Session3_24 June 2020; User ID: 2557, female, 55–64)	“I really appreciate being in this group. As I read all of your stories I feel less alone.” (Advanced_CancerSession1_30 April 2021; User ID: 2338, female, 35–44)	“Scary though[t] but we are in it together.” (Advanced_Cancer_Session1_30 April 2021; User ID: 3197, male, 45–54)
	Providing Support to Others	“*****- it has been somewhat helpful for me to look in to some supports around death. It may be very early but it helps to feel like I have my ducks in a row. I’ve looked into Greensleeves, MAID etc.” (Advanced_Cancer _Session7_11 June 2021; User ID: 3034, female, 55–64)	“******- that is totally fair and everyone’s risk level is different. You may change in the future and not now so that is okay.” (Advanced_Cancer_Session3_ 19 Aug 2020; User ID: 2799, unknown, unknown)	“Unfortunately I know a lot of people in the cancer world, friends and two sisters, so many around me can relate and support. I tend to help support others who are going through non cancer problems… pain is pain.”(Advanced_Cancer_Session6_22 July 2020; User ID: 2877, female, 45–54)	“I always find comfort in supporting others who are going through the same thing.” (Advanced_Cancer_Session4_21 May 2021; User ID: 3215, female, 25–34)					
Existential Experiences	Living with Diagnosis for an Extended Amount of Time	“Three years stable and feeling blessed.” (Advance_Cancer_Session1_5 Aug 2020; User ID: 2877, female, 45–54)	“I never got angry when first diagnosed 11 years ago. I am still not angry as it won’t do me any good. This is what I have been dealt with and so this is what I have to do. And i would rather have this than my mom or sister. I can do this, and I am.” (Advanced_Cancer_Session1_30 April 2021; User ID: 3212, female, 55–64)							
	Legacy and Death Preparations	“I have taken over someone else’s legacy and am adding to it and then will pass it on to the next person when the time comes.” (Advanced_Cancer_Session6_4 June 2021; User ID: 3203, female,55–64)	“I’d like to leave behind something that may help others in similar situations. Yes, I am also beginning a big purge project and try to reorganize my stuff.” (Advanced_Cancer_Session6_4 June 2021; User ID: 2987, female 65+)	“I have started an EOL plan… it’s hard but needs to be done.” (Advanced_Cancer_Session4_21 May 2021; User ID: 3215, female, 25–34)						
	Appreciating Life	“I am thankful that I am still here enjoying life. I just had another friend lose her battle on wednesday and it makes me appreciate life so much!” (Advanced_Cancer_Session3_14 May 2021; User ID: 3203, female, 55–64)	“I can’t work anymore, so I’m trying to see myself as retired, but with some health problems.” “Advanced _Cancer_Session2_12 August 2020; User ID: 2417, female, 55–64)	“Will always be in treatment but thankful and happy to be here. Side effects and all.” (Advanced_Cancer_Session1_30 April 2021; User ID: 3212, female,55–64)	“as I can’t change the past - I’ve worked hard at releasing the negative energy that it brings.that’s (sic) true Sue—I enjoy ‘everyday\\” and think of it as a [gift].” (Advanced_Cancer_Session7_16 September 2020; User ID: 2557, female, 55–64)	“[Cancer] has definitely allowed me to experience life more fully.” (Advanced_Cancer_Session7_11 June 2021; User ID: 3203, female, 55–64)				

Note: To protect the identity of participants, asterisks were used to conceal any personal information, such as names, and locations.

## Data Availability

The data presented in this study are available on request from the corresponding author. The data are not publicly available due to participant privacy and confidentiality.

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
