# Peer review of "Pathways to Acceptance in Participants of Advanced Cancer Online Support Groups"

_medicina, 2021, doi:10.3390/medicina57111168_

Round 1

Reviewer 1 Report

This is an original work about a prospective research: exploring the factors that interfere with the disease acceptance in patients with advanced cancer. The methodology is adequate, based on material taken from online support group sessions. The presentation of the text and the division of sections follow the academic format. The subject has relevance to the medical field, and the results can be used by other groups.
The following demands are just to adapt the manuscript to the quality standard of this journal:

Please clarify more about the Informed Consent Form. Was it verbal or did the participant click agreeing with a text? Was it clear that participation in the study was voluntary, with the right to deny sharing the information? Was it clear that he/she could stop participating at any time, without negative consequences for his/her follow-up? If all answers are positive, please insert these sentences into the text.

Please detail the process of the casuistry composition. Were there exclusion criteria to be part of the sample? Were the participants those who were being followed-up in which period (months of 2020 or 2021)? In this period, how many patients in total were in follow-up (to have the acceptance percentage)?

Author Response

Dear Reviewer,

Thank you for reviewing the manuscript. I have included my point-by-point responses in red to your comments in black. 

1. Please clarify more about the Informed Consent Form. Was it verbal or did the participant click agreeing with a text? Was it clear that participation in the study was voluntary, with the right to deny sharing the information? Was it clear that he/she could stop participating at any time, without negative consequences for his/her follow-up? If all answers are positive, please insert these sentences into the text.

Response: Thank you for bringing this comment to our attention. Group members were asked for their consent using an online form prior to commencing the online support groups (OSG) by a program coordinator over the phone. These individuals were also reassured that participation was voluntary and if they did not wish to consent, they could join an OSG that was not going to be analyzed. They were also informed that if they wished to withdraw their consent at any time, they would be able to switch to another OSG that was not being analyzed, without any consequence. This has been clarified in the Methods section of the manuscript. 

2. Please detail the process of the casuistry composition. Were there exclusion criteria to be part of the sample? 

Response: The only inclusion/exclusion criterion was for individuals to be in the advanced stage of their disease. We included all results from consenting participants that were engaged in an OSG for advanced cancer patients. There was no difference in treatment between the two groups we analyzed, only that they occurred at different times to fit participants’ schedules. 

Individuals who did not fit this criterion (i.e. individuals that were still in active treatment), were still able to join other OSGs appropriate for their stage of disease; however based on the nature of our investigation of advanced cancer populations, we did not analyze other patient populations.

 3. Were the participants those who were being followed-up in which period (months of 2020 or 2021)? In this period, how many patients in total were in follow-up (to have the acceptance percentage)?  

Response: The participant data were analyzed only in the sessions that they engaged in. If participants stopped coming, we did not follow up with them, as their cessation may have been due to the progression of their disease or other personal reasons, of which we did not systematically keep track. Given the qualitative nature of the study, we did not plan to do a formal longitudinal study with follow ups. The lack of formal follow-up of dropouts has been included as a limitation of the study in the Limitations section.

Thank you so much for all the insightful comments and the opportunity for us to improve the manuscript. I hope you find our responses satisfactory. 

Sincerely,

Christina Pereira

Reviewer 2 Report

Thank you for the opportunity to review this manuscript.  The overall research and writing is very well done.  Yalom's  therapeutic factors could be described and explained for the reader earlier in the manuscript.  Although I am familiar with Yalom's existential psychodynamics or givens (death, freedom, isolation and meaninglessness), I did not see them described until later in the manuscript. 

Lines 137-138 refer to previously developed theories of acceptance.  Which theories?   

Is there a way to link the result themes and subthemes with each of the four givens? In this manuscript, the authors did a good job with Yalom's existential factors which provide framework for the study.  To some extent, I am not able to catch the full essence of Yalom's factors.  They seem to get lost in the main concept of acceptance. 

Since death is one of the concepts written about in this manuscript and the participants "acceptance", would the authors consider briefly linking or differentiating from Kubler Ross? 

Future Directions.  The topic of Artificial Intelligence comes as a surprise.  It is not mentioned anywhere else in the manuscript.  

Well done manuscript.  Thank you again.  

Author Response

Dear Reviewer, 

Thank you for taking the time to review this manuscript. We have outlined our responses in red to your comments in black below:

1. Thank you for the opportunity to review this manuscript.  The overall research and writing is very well done.  Yalom's  therapeutic factors could be described and explained for the reader earlier in the manuscript.  Although I am familiar with Yalom's existential psychodynamics or givens (death, freedom, isolation and meaninglessness), I did not see them described until later in the manuscript. 

Response: Thank you for pointing out this concern. We have amended the introduction section so that it explicitly called attention to Yalom’s theory and how it is related to the current research

2. Lines 137-138 refer to previously developed theories of acceptance.  Which theories?   

Response: Thank you for bringing this to our attention. We were referring to the acceptance-based literature that had been cited in the previous paragraph. To avoid any confusion, we have repeated the citations so that it is clear which theories we were referring to. 

3. Is there a way to link the result themes and subthemes with each of the four givens?

Response: We have added additional details to the discussion to provide a more comprehensive link between the themes and sub-themes and how this relates to Yalom’s theory.

4. In this manuscript, the authors did a good job with Yalom's existential factors which provide framework for the study.  To some extent, I am not able to catch the full essence of Yalom's factors.  They seem to get lost in the main concept of acceptance. 

Response: Thank you for pointing out this confusion. We have added additional information to the introduction to further elaborate on Yalom’s theory of existential factors

5. Since death is one of the concepts written about in this manuscript and the participants "acceptance", would the authors consider briefly linking or differentiating from Kubler Ross? 

Response: Thank you for this additional insight. We have included the theory by Kubler Ross and compared it to that of Yalom in the Future Direction section. 

6. Future Directions.  The topic of Artificial Intelligence comes as a surprise.  It is not mentioned anywhere else in the manuscript

Response: Thank you for this comment. The topic of Artificial Intelligence has been removed from the Future Directions section.

Thank you so much for all the insightful comments and the opportunity for us to improve the manuscript. I hope you find our responses satisfactory. 

Sincerely,

Christina Pereira